# Association of Body Mass Index with Risk of Household Catastrophic Health Expenditure in China: A Population-Based Cohort Study

**DOI:** 10.3390/nu14194014

**Published:** 2022-09-27

**Authors:** Yaping Wang, Min Liu, Jue Liu

**Affiliations:** 1Department of Epidemiology and Biostatistics, School of Public Health, Peking University, Beijing 100191, China; 2Institute for Global Health and Development, Peking University, Beijing 100871, China; 3Key Laboratory of Reproductive Health, National Health and Family Planning Commission of the People’s Republic of China, Beijing 100191, China; 4Global Center for Infectious Disease and Policy Research & Global Health and Infectious Diseases Group, Peking University, Beijing 100191, China

**Keywords:** catastrophic health expenditure, body mass index, universal health coverage

## Abstract

Catastrophic health expenditure (CHE) is a major obstacle to achieving universal health coverage, and body mass index (BMI) is linked to both health and economy. We aimed to explore the association of BMI with the risk of CHE to provide advice for reducing CHE. We used national cohort data from the China Family Panel Studies, which comprised 33,598 individuals (14,607 households) from 25 provinces between 2010 to 2018. We used multivariate Cox proportional hazard models to estimate adjusted hazard ratios (aHRs) and 95% confident interval (CI) for CHE in participants at underweight, overweight, and obesity, compared with those at normal weight. Restricted cubic splines were employed to model the association of continuous BMI scale with risk of CHE. We found that families with female household heads at underweight had a 42% higher risk of CHE (aHR = 1.42, 95%CI: 1.16–1.75), and those at overweight had a 26% increased risk of CHE (aHR = 1.26, 95%CI: 1.09–1.47), compared with those at normal weight. A weak U-shaped curve for the association of continuous BMI with risk of CHE in female-headed households (*p* for non-linear = 0.0008) was observed, which was not significant in male-headed households (*p* for non-linear = 0.8725). In female-headed households, underweight and overweight BMI are connected with a higher risk of CHE. Concerted efforts should be made to keep a normal BMI to prevent CHE.

## 1. Introduction

Universal health coverage (UHC), which was embedded in the Sustainable Development Goals (SDGs) agreed by the United Nations in 2015,aimed to achieve that “all people have access to the health services they need, when and where they need them, without financial hardship” by 2030 [1]. To monitor progress towards UHC, catastrophic health expenditure (CHE), defined as household expenditure on health-care services exceeds a specific threshold of household total expenditure or income, was recommended as an indicator [2]. According to the World Health Organization (WHO), globally in 2017, almost 1.4 billion people occurred financial hardship due to out-of-pocket (OOP) health payments, of whom nearly 1 billion people were pushed into extreme poverty [3]. Hence, there is still a long way to achieve UHC.

A great number of factors can affect the incidence of CHE, the main of which can be divided into health-related and economic factors [4,5,6]. Poor health conditions, obviously, can increase the demand and usage of medical care, further leading to high costs in health-care and finally triggering CHE. Zhao et al. [5] found that after adjusting for sociodemographic confounders, physical multimorbidity was significantly associated with an increased number of outpatient visits (odds ratio = 1.29, 95% confident interval (CI): 1.27–1.31), inpatient days spent (odds ratio = 1.38, 95%CI: 1.35–1.41), and likelihood of CHE (odds ratio = 1.29, 95%CI: 1.26–1.32). As for economic factors, a series of studies revealed that families with lower socioeconomic status, such as lower family income, lower education level, and higher unemployment, were more likely to incur CHE [7,8,9]. Therefore, in order to eliminate CHE and move to UHC, one of the key points is finding ways to light the heavy health-economic burden.

Body mass index is an indicator for overweight and obesity, which is associated with lots of health problems. A system review and meta-analysis reported that the all-cause mortality increased approximately log-linearly with body mass index (BMI) for a hazard ratio (HR) of 1.26 to 2.44 per 5 kg/m^2^ units higher BMI in Europe, North America, east Asia, Australia, and New Zealand [10]. The adverse impact of BMI on health may be presented after a long period and modified by growth. Blond et al. [11] found that compared with children in the lowest BMI level, those in several higher BMI trajectories were associated with higher mean waist circumference, lower high-density lipoprotein (HDL), and higher risk of diabetes in adulthood, while these associations can be reversed when adjusting for adult BMI. In addition, lower BMI is also connected with some health problems. Qu and colleagues reported that midlife underweight and late-life underweight conferred 1.39- and 1.64-fold excess risk for cognitive impairment and dementia, and this association was also found in other studies [12,13]. Besides the close relation with health, as a result of interactions between inheritance, environmental, socioeconomic, and life experience, BMI is also identified as a socioeconomic indicator [14]. Compared with individuals at normal BMI, the median increases of mean total annual healthcare costs were 12% for overweight and 36% for obesity individuals [15].

Overall, BMI plays a role in individual’s health condition and medical costs, which may cause a connection to CHE. However, current studies are mainly about the national financial burden of obesity or the effect of economic status on obesity, rarely about association of BMI with risk of household CHE [16,17]. This study aimed to explore the relationship between BMI and risk of CHE in a national longitudinal study based on the China Family Panel Studies, and further wanted to provide more ways to reduce the incidence of CHE.

## 2. Materials and Methods

### 2.1. Study Design and Participants

The China Family Panel Studies (CFPS), which was implemented by the Institute of Social Science Survey (ISSS) of Peking University, is a nationally-representative follow-up interview study covering 25 out of 31 provinces/municipalities in China, and representing nearly 94.5% of the total population [18]. The CFPS is intended to collect individual, family, and community-level data every two years. A baseline survey was conducted between April 2010 and February 2011, and follow-up data of 2012, 2014, 2016, and 2018 waves were available to be downloaded from the CFPS official website (http://www.isss.pku.edu.cn/cfps/ accessed on 1 July 2022). The study was approved by the Peking University Biomedical Ethics Review Committee (protocol code IRB00001052-14010). All participants sighed informed consent before enrolled.

In this study, we used all waves data interested from adult questionnaire and family questionnaire of the CFPS. Baseline survey interviewed 35,720 adults (aged ≥ 16) and 14,607 households. Additionally, among the 14,607 households totally interviewed, 11,634, 11,238, 10,540, and 9698 were successfully tracked in 2012, 2014, 2016, and 2018. In our study, households who incurred CHE at baseline (*n* = 2146), were lost to follow-up without any information of CHE (*n* = 1245), and had missing data of baseline characteristics (*n* = 31) were excluded. Finally, a total of 11,185 households (heads) were included in this study (Figure 1).

### 2.2. Procedure

The CFPS used multistage probability proportional to size sampling (PPS) to select participants throughout three stages: (i) the primary sampling unit (PSU) was either an administrative district (in urban areas) or a county (in rural areas); (ii) the second sampling unit was either a neighborhood community (urban areas) or an administrative village (rural areas); and (iii) the third sampling unit was the household [19]. Participants enrolled were interviewed face-to-face by well-trained local interviewers with a series of structured questionnaires aided by computer-assisted personal interviewing technology. The contents of these questionnaires were thorough and comprehensive as the design team of the CFPS learned from the approaches and experiences of earlier successful research programs, such as the Panel Study of Income Dynamics, the National Longitudinal Surveys of Youth, the Health and Retirement Study, and so on [19]. The major contents of interest in our study included demographic characteristics (gender, age, marital status, education, medical insurance, and so on) and health-related characteristics (weight, height, self-reported health status, diagnosed chronic diseases in the past 6 months, the utilization of health services, smoking, drinking, and so on) in the adult questionnaire, and socioeconomic characteristics (residence, household income, household expenditures and so on) in the family questionnaire.

We conducted a population-based cohort study based on the CFPS. First, we calculated baseline BMI as weight (kg)/height^2^ (m^2^). Then, based on underweight defined by WHO and Chinese criteria for overweight and obesity, we divided participants into four groups according to their BMI values: people at normal weight (18.5 kg/m^2^ ≤ BMI < 24 kg/m^2^, unexposed group); people at underweight (BMI < 18.5 kg/m^2^, exposed group 1); people at overweight (24 kg/m^2^ ≤ BMI < 28 kg/m^2^, exposed group 2); and people at obesity (BMI ≥ 28 kg/m^2^, exposed group 3) [20,21]. Finally, we followed them up until they had CHE or the interview ended.

### 2.3. Outcome

In this study, a household with CHE was defined as household OOP medical expenditure exceeded 40% of household’s capacity to pay (defined as total household expenditure minus household food expenditure) [22]. In the CFPS, a family member was defined by marriage, blood, or adoptive relationship and an on-going economic tie [19]. The household head was identified as the key decision maker when household faced important matters and decisions. The household OOP health payments were measured as the expenditure on medical care, including outpatient and inpatient care and other types of healthcare, of all family members excluding reimbursed expending in last year. The household food expenditure was estimated as the monthly meal expenses multiplied by 12, and the total household expenditure in last year was calculated as the sum of monthly daily expenditures (food, daily used commodities and necessities, transportation, and so on) multiplied by 12 plus yearly special expenditures (electricity, medical care, clothing, and so on).

### 2.4. Covariates

Covariates in this study include: (i) demographic characteristics: gender (male, female), age group (16–39, 40–49, 50–49, ≥60), marital status (married/partnered, other), education (illiterate/semiliterate, primary school, middle school, high school and above), and insurance (without any insurance, urban employee basic medical insurance (UEBMI), urban resident basic medical insurance (URBMI), new rural cooperative medical scheme (NRCMS), other); (ii) health-related characteristics: self-reported health (good, medium, poor), chronic diseases (yes, no), outpatient services (yes, no), inpatient services (yes, no), current smoking (yes, no), and drinking (yes, no); (iii) socioeconomic characteristics: residence (urban, rural), family economic level (four classes), family size (1–2, 3–4, ≥5), and socioeconomic development level (four classes).

The outpatient services were obtained from question “Whether have outpatient care in the past two weeks”, and the inpatient services were measured by “How many times were you hospitalized due to illness last year”. The family economic level was classified by the quartiles of household annual income (lowest: <14,420 CNY; lower: 14,420–25,530 CNY; higher: 25,531–43,704 CNY; highest: ≥43,705 CNY). Socioeconomic development level in this study was identified by the quartiles of 2010 per capita gross regional product (GRP, lowest: <21,182 CNY; lower: 21,182–27,132 CNY; higher: 27,133–42,354 CNY; highest: ≥42,355 CNY), which were obtained from the 2010 China Statistical Yearbook [23].

### 2.5. Statistical Analysis

The baseline characteristics of the participants were described as mean ± standard deviation (SD) for continuous variables or frequencies and percentages for categorical variables. The Person χ^2^ test was used to compare the difference in distributions of characteristics according to BMI group.

We calculated the incidence rates (number of events divided by accumulated person-month) and used the univariate and multivariate Cox proportional hazard models to estimate the HRs and 95%CIs of CHE among participants at underweight, overweight, and obesity, compared with those at normal weight. Time to CHE event was defined as the period from the month of baseline survey to the month when household head first reported OOP medical costs exceeding 40% of non-food expenditure. Additionally, the censored time was calculated as the period from the baseline survey month to the last available wave survey month for those households who did not have CHE events until the investigation ended or who were recorded as without CHE in this survey wave but were lost to follow up in the next survey wave.

To examine the robustness of our findings, we performed three sensitivity analyses. First, we established three models adjusted for different covariates to estimate adjusted hazard ratios (aHRs) and their CIs. In model 1, we adjusted demographic characteristics, including gender (male, female), age group (16–39, 40–49, 50–49, ≥60), marital status (married/partnered, other), education (illiterate/semiliterate, primary school, middle school, high school and above), and insurance (none, UEBMI, URBMI, NRCMS, other) based on univariate model. In model 2, we further included health-related characteristics, including self-reported health (good, medium, poor), chronic diseases (yes, no), outpatient services (yes, no), inpatient services (yes, no), current smoking (yes, no), and drinking (yes, no). In model 3 (final fully-adjusted model), besides those factors included in model 2, we also adjusted for socioeconomic characteristics, including residence (urban, rural), family economic level (lowest, lower, higher, highest), family size (1–2, 3–4, ≥5), and socioeconomic development level (lowest, lower, higher, highest). Second, we transferred the categorical variables age group and family economic level into continuous variables and conducted the same analysis in final model. Third, we used nightlight intensity (also divided into four classes according to the quartiles of 2010 province-level mean nightlight intensity [24]) to indicate socioeconomic development rather than GRP in final model.

Furthermore, the analysis was stratified by age group, insurance, chronic diseases, self-reported health, current smoking and drinking, outpatient services, inpatient services, residence, socioeconomic development level, and family economic level in the fully adjusted model with the stratified variables removed. We also used restricted cubic splines with knots at 5th, 35th, 65th, and 95th percentiles to flexibly model the association of the continuous scale of BMI with CHE incidence after adjusting covariates. Additionally, we tested the potential non-liner association by using a likelihood ratio test to compare the model with only a liner term against the model with linear and cubic spline terms. Because it is widely known that male and female have different body compositions, such as lean mass and fat mass, these analyses were stratified by gender.

All of the data were analyzed in R 4.2.1 (R Core Team, Vienna, Austria). Two-side *p*-value less than 0.05 was considered to be significant.

## 3. Results

### 3.1. Baseline Characteristics

Among all of 11,185 participants, the mean (SD) age was 48.3 (12.9) years, and 7864 (70.3%) were male, 3321 (29.7%) were female, 9836 (87.9%) were married or partnered, and only 1394 (12.5%) participants did not have any medical insurance (Table 1). At baseline, 6833 (61.1%), 831 (7.4%), 2825 (25.3%), and 696 (6.2%) participants were in the normal weight, underweight, overweight, and obesity group, respectively. Overall, except inpatient services, the distribution of baseline characteristics among participants in four BMI groups was significant different (all *p*-value < 0.05).

### 3.2. Risk of CHE

During a median (interquartile range) of 95 (50–97) person-month of follow-up, a total 3275 households incurred CHE with an incidence rate of 3.85 per 1000 person-month. The incident number (incidence rate) of CHE was 1968 (3.77 per 1000 person-month) for participants at normal weight, 298 (5.08 per 1000 person-month) for those at underweight, 822 (3.82 per 1000 person-month) for those at overweight, and 187 (3.47 per 1000 person-month) for those at obesity.

In total, the significant association of BMI with a risk of CHE was only observed in participants at underweight, compared with those at normal weight (Table 2). In the unadjusted model, compared with individuals at normal weight, those at underweight had a 38% increased risk of CHE (crude hazard ratio = 1.38, 95%CI: 1.22–1.55). Additionally, in fully adjusted model (Model 3), participants at underweight had a 15% higher risk of CHE than those at normal weight (Table 2 and Appendix A, aHR = 1.15, 95%CI: 1.02–1.31).

In fully adjusted model (Model 3) for female participants, compared with those at normal weight, individuals at underweight (aHR = 1.42, 95%CI: 1.16–1.75) and overweight (aHR = 1.26, 95%CI: 1.09–1.47) had a 42% and 26% increased risk of CHE, respectively, while there were no significant connections between BMI and CHE observed in male (Table 2).

In Figure 2, we further used restricted cubic splines to flexibly model and visualize the relationship between continuous scale of BMI and risk of CHE by gender. Though a nonlinear relation was significant only in female individuals, the risk of CHE increased when BMI decreased below the medians for the total and for the female heads. Additionally, just above the median BMI, a slight increased risk of CHE with higher BMI was observed in female.

### 3.3. Sensitivity Analyses and Subgroup Analyses

In the sensitivity analyses, the associations of underweight and overweight with a risk of CHE was robust with different characteristics adjusted (demographic characteristics, health-related characteristics, socioeconomic characteristics); or categorical variables age group and family economic level changed into continuous variables; or socioeconomic development level indicated by nightlight intensity rather than GRP in female individuals (Appendix A).

In subgroup analyses, the associations of underweight and overweight with a risk of CHE in female were significant in individuals aged 50–59 years, NRCMS, not currently smoking, having undergone outpatient services, and not having undergone inpatient services (Table 3, all *p*-value < 0.05).

## 4. Discussions

As we know, there is a bi-directional relationship between health and poverty, and CHE is a good example for interpreting this phenomenon: poor health conditions cause higher medical costs, which may push a household into poverty if the economic burden is unaffordable, and further leads to worse health. In order to reduce CHE incidence and move further towards UHC, more focus should be paid to detecting and resolving the factors of CHE. In this study, we found that families with female household heads at underweight and overweight had a higher risk of CHE than those at normal weight. Integrated efforts should be made to maintain a normal BMI to not only prevent health problems, but also to avoid CHE incidents.

Our results discovered that female people at underweight had a 42% higher risk of CHE (aHR = 1.42, 95%CI: 1.16–1.75) and those at overweight had a 26% increased risk of CHE (aHR = 1.26, 95%CI: 1.09–1.47), compared with those at normal weight. From the definition of CHE, it is not difficult to find that there are two main causes to trigger CHE, high OOP payments for healthcare and large non-food household expenditures or low family income. The former appears to be somewhat connected with health effects and medical insurance, and the latter is somewhat relevant to economic factors. It is common knowledge that higher BMI is related to worse health [25,26,27]. A recently published article revealed that high BMI was the third risk factors contributing to the global cancer burden of age-standardized disability-adjusted life years rates (133.9 per 100,000 person-years) [25]. Another Mendelian randomization study also demonstrated that higher BMI was associated with increased risk of most cardiovascular conditions [26]. Meanwhile, there is also a close linkage between lower BMI and poor health [28]. Additionally, a great number of studies revealed a U-shaped curve between BMI and some health outcomes, like all-cause mortality and heart diseases [29,30]. That is to say, both lower BMI and higher BMI can have a negative impact on health, which may increase the OOP expense for healthcare.

Nevertheless, in this study, the stronger effect on CHE of underweight, rather than overweight, may be more explained by economic connections. A study, focusing on the contribution of socioeconomic factors to the variation of BMI in 59-low-income and middle-income countries, found that women in the wealthiest group had a 2.3 kg/m^2^ higher BMI than those in the poorest group [31]. Similarly, Razak et al. [32] revealed that the prevalence of BMI lower than 16 kg/m^2^ was associated with poverty and low education levels, and this prevalence did not increase over time in most countries studied. Since lower BMI is relevant to poverty and, furthermore, linked to poverty-related diseases [33], the association of poverty and health, as well as lower socioeconomic level with risk of CHE is widely verified [7,8,9], and it cannot be difficult to understand that underweight BMI is related to higher incidence of CHE.

In the gender-stratified restricted cubic splines analyses, we found that a light lean U-shaped curve for the association of continuous BMI with risk of CHE in female individuals (*p* for non-linear = 0.0008), which was not significant in male individuals (*p* for nonlinear = 0.8725). Overall, compared with male-headed households, female-headed households were more likely to have a lower socioeconomic status, such as having a smaller family size, living in rural areas, not having enough food to eat, and so on, which made them more vulnerable to financial problems [34]. Several systematic-reviews and meta-analyses discovered that the pooled prevalence of food insecurity among female-headed households is 66.1% (95%CI: 54.61–77.60), with a 40–94% higher risk of developing food insecurity than male-headed households [35,36,37]. Insecure food situations, including insufficient food, unsafe food resources, uncertainty about the access to food, and the experience of hunger, is highly related with food patterns and unhealthy BMI [38,39]. The explanation for the U-shaped curve for continuous BMI with risk of CHE in female individuals, on the one hand, is that poor families often do not receive enough food and are more likely to eat contaminated food, which are factors linked to lower BMI and worse health. On the other hand, with financial strain, most households in poverty will often turn to low energy-cost but high energy-density food, like those foods of grains, added sugars, and fats, which is related to fast BMI growth [40,41].

In order to cut the apparent link between BMI and CHE, the main methods are health promotion and financial protection. First, the potential cost of healthcare should be reduced by minimizing or delaying the onset of diseases, especially chronic and severe disease that requires long-term care. Second, a complete medical security system should be established and provided for those who are sick, and special targeted protection measures should be implemented for those at high risk of a CHE event. For example, the government of China launched catastrophic medical insurance (critical illness insurance) in 2012 and it was implemented nationwide in 2016 after city-based testing, which aimed to reimburse patients whose OOP health expenditure exceeding a predetermined basic medical insurance level [42]. Third, for key populations, such as female-headed households and families with extreme poverty, interventions regarding the protection of basic living and primary health services should be expanded.

Our study is a useful design to evaluate the effect of BMI on household CHE. However, there are still several limitations in our study. First, data for calculating BMI and CHE, including weight, height, family medical expenditure, family total expenditure, and family food expenditure were mainly based on self-reported answers, which could be affected by recalling bias. Second, as some family members were not always living at home, their real expenditures in last year may be unclear to access from household heads. Third, though our study adjusted a range of potential confounders, there still are some extant unadjusted confounders. Fourth, as the CFPS is designed for the Chinese population and some observations were excluded in our study, the representation for the global population is limited, which needs further research.

## 5. Conclusions

Our study found that underweight and overweight BMI are associated with a higher risk of CHE incidence, compared with normal BMI, in female-headed households. Additionally, a weak U-shaped curve was observed between the continuous scale of BMI and CHE incidence among female-headed households. Concerted efforts should be made to encourage the public to maintain normal weight. Moreover, to receive UHC by 2030, timely preventive interventions concerning CHE need to be implemented among the key populations.

## Figures and Tables

**Figure 1 nutrients-14-04014-f001:**
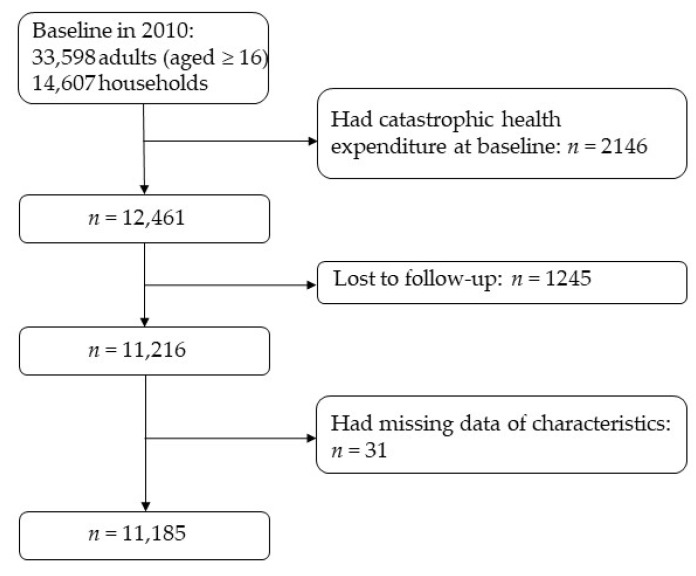
Flowchart of the study population.

**Figure 2 nutrients-14-04014-f002:**
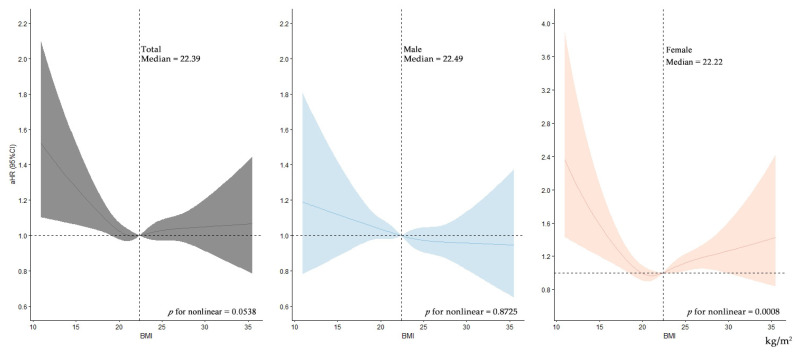
Restricted cubic splines analyses for association between BMI and risk of CHE stratified by gender. Notes: BMI: body mass index; aHR: adjusted hazard ratio; CI: confident interval. aHR (95%CI) was adjusted for all covariates, including demographic characteristics (gender, age group, education, marital status, and insurance), health-related characteristics (self-reported health, current smoking, drinking, chronic disease, outpatient and inpatient services), and socioeconomic characteristics (residence, family economic level, family size, and socioeconomic development level).

**Table 1 nutrients-14-04014-t001:** Distribution of baseline characteristics among participants with different BMI group.

Characteristics	Total (*n* = 11,185, %)	BMI Group				χ^2^	*p*-Value
		Normal (*n* = 6833, %)	Underweight (*n* = 831, %)	Overweight (*n* = 2825, %)	Obesity (*n* = 696, %)		
Gender						36.359	<0.001
Male	7864 (70.3)	4859 (71.1)	508 (61.1)	2000 (70.8)	497 (71.4)		
Female	3321 (29.7)	1974 (28.9)	323 (38.9)	825 (29.2)	199 (28.6)		
Age group						150.112	<0.001
16–39	2843 (25.4)	1777 (26.0)	223 (26.8)	656 (23.2)	187 (26.9)		
40–49	3452 (30.9)	2134 (31.2)	159 (19.1)	937 (33.2)	222 (31.9)		
50–59	2689 (24.0)	1602 (23.4)	173 (20.8)	746 (26.4)	168 (24.1)		
≥60	2201 (19.7)	1320 (19.3)	276 (33.2)	486 (17.2)	119 (17.1)		
Marital status						172.815	<0.001
Married/partnered	9836 (87.9)	6013 (88.0)	619 (74.5)	2571 (91.0)	633 (90.9)		
Other	1349 (12.1)	820 (12.0)	212 (25.5)	254 (9.0)	63 (9.1)		
Education						229.064	<0.001
Illiterate/semiliterate	2732 (24.4)	1749 (25.6)	316 (38.0)	546 (19.3)	121 (17.4)		
Primary school	2649 (23.7)	1698 (24.9)	204 (24.5)	604 (21.4)	143 (20.5)		
Middle school	3461 (30.9)	2069 (30.3)	207 (24.9)	951 (33.7)	234 (33.6)		
High school and above	2343 (20.9)	1317 (19.3)	104 (12.5)	724 (25.6)	198 (28.4)		
Insurance						241.867	<0.001
None	1394 (12.5)	833 (12.2)	121 (14.6)	346 (12.2)	94 (13.5)		
UEBMI	1354 (12.1)	692 (10.1)	43 (5.2)	471 (16.7)	148 (21.3)		
URBMI	757 (6.8)	426 (6.2)	39 (4.7)	237 (8.4)	55 (7.9)		
NRCMS	6685 (59.8)	4318 (63.2)	532 (64.0)	1506 (53.3)	329 (47.3)		
Other	995 (8.9)	564 (8.3)	96 (11.6)	265 (9.4)	70 (10.1)		
Self-reported health						102.954	<0.001
Good	5279 (47.2)	3279 (48.0)	296 (35.6)	1395 (49.4)	309 (44.4)		
Medium	4237 (37.9)	2557 (37.4)	324 (39.0)	1077 (38.1)	279 (40.1)		
Poor	1669 (14.9)	997 (14.6)	211 (25.4)	353 (12.5)	108 (15.5)		
Outpatient services						17.269	0.002
No	9113 (81.5)	5569 (81.5)	636 (76.5)	2342 (82.9)	566 (81.3)		
Yes	2072 (18.5)	1264 (18.5)	195 (23.5)	483 (17.1)	130 (18.7)		
Inpatient services							
No	10,512 (94.0)	6432 (94.1)	779 (93.7)	2652 (93.9)	649 (93.2)	1.075	0.783
Yes	673 (6.0)	401 (5.9)	52 (6.3)	173 (6.1)	47 (6.8)		
Chronic diseases						38.440	<0.001
No	9550 (85.4)	5940 (86.9)	696 (83.8)	2355 (83.4)	559 (80.3)		
Yes	1635 (14.6)	893 (13.1)	135 (16.2)	470 (16.6)	137 (19.7)		
Smoking						38.664	<0.001
No	6053 (54.1)	3543 (51.9)	460 (55.4)	1649 (58.4)	401 (57.6)		
Yes	5132 (45.9)	3290 (48.1)	371 (44.6)	1176 (41.6)	295 (42.4)		
Drinking						18.624	<0.001
No	8364 (74.8)	5091 (74.5)	671 (80.7)	2099 (74.3)	503 (72.3)		
Yes	2821 (25.2)	1742 (25.5)	160 (19.3)	726 (25.7)	193 (27.7)		
Residence						238.227	<0.001
urban	5084 (45.5)	2828 (41.4)	292 (35.1)	1548 (54.8)	416 (59.8)		
rural	6101 (54.5)	4005 (58.6)	539 (64.9)	1277 (45.2)	280 (40.2)		
Family size						75.962	<0.001
1–2	2165 (19.4)	1245 (18.2)	191 (23.0)	592 (21.0)	137 (19.7)		
3–4	5431 (48.6)	3249 (47.5)	344 (41.4)	1457 (51.6)	381 (54.7)		
≥5	3589 (32.1)	2339 (34.2)	296 (35.6)	776 (27.5)	178 (25.6)		
Family economic level						185.968	<0.001
Lowest	2941 (26.3)	1863 (27.3)	325 (39.1)	625 (22.1)	128 (18.4)		
Lower	2567 (23.0)	1642 (24.0)	167 (20.1)	616 (21.8)	142 (20.4)		
Higher	3164 (28.3)	1910 (28.0)	222 (26.7)	824 (29.2)	208 (29.9)		
Highest	2513 (22.5)	1418 (20.8)	117 (14.1)	760 (26.9)	218 (31.3)		
Socioeconomic development level						238.177	<0.001
Lowest	2358 (21.1)	1615 (23.6)	268 (32.3)	401 (14.2)	74 (10.6)		
Lower	3234 (28.9)	1930 (28.2)	211 (25.4)	907 (32.1)	186 (26.7)		
Higher	2015 (18.0)	1201 (17.6)	111 (13.4)	528 (18.7)	175 (25.1)		
Highest	3578 (32.0)	2087 (30.5)	241 (29.0)	989 (35.0)	261 (37.5)		

Notes: BMI: body mass index; UEBMI: urban employee basic medical insurance; URBMI: urban resident basic medical insurance; NRCMS: new rural cooperative medical scheme.

**Table 2 nutrients-14-04014-t002:** Univariate and multivariate Cox proportional hazard analyses for association of BMI with risk of CHE stratified by gender.

BMI Groups	Events/Incidence Rate *	Univariate Model		Model 1		Model 2		Model 3	
		cHR (95%CI)	*p*-Value	aHR (95%CI)	*p*-Value	aHR (95%CI)	*p*-Value	aHR (95%CI)	*p*-Value
Total	3275/3.85								
Normal	1968/3.77	Ref.	—	Ref.	—	Ref.	—	Ref.	—
Underweight	298/5.08	1.37 (1.22–1.55)	<0.001	1.18 (1.04–1.34)	0.008	1.14 (1.01–1.29)	0.036	1.15 (1.02–1.31)	0.023
Overweight	822/3.82	1.01 (0.93–1.10)	0.761	1.06 (0.98–1.15)	0.162	1.06 (0.97–1.15)	0.177	1.05 (0.97–1.15)	0.210
Obesity	187/3.47	0.92 (0.79–1.07)	0.293	0.98 (0.85–1.14)	0.840	0.98 (0.84–1.13)	0.750	0.98 (0.84–1.14)	0.747
Male	2288/3.79								
Normal	1449/3.87	Ref.	—	Ref.	—	Ref.	—	Ref.	—
Underweight	180/4.94	1.29 (1.11–1.51)	0.001	1.07 (0.92–1.26)	0.368	1.03 (0.88–1.21)	0.706	1.05 (0.90–1.23)	0.547
Overweight	543/3.53	0.90 (0.82–1.00)	0.045	0.98 (0.88–1.08)	0.649	0.98 (0.89–1.08)	0.702	0.97 (0.88–1.07)	0.570
Obesity	116/2.99	0.77 (0.64–0.93)	0.007	0.89 (0.73–1.07)	0.210	0.88 (0.73–1.07)	0.196	0.87 (0.72–1.05)	0.152
Female	987/4.00								
Normal	519/3.49	Ref.	—	Ref.	—	Ref.	—	Ref.	—
Underweight	118/5.32	1.57 (1.28–1.91)	<0.001	1.43 (1.17–1.75)	0.001	1.42 (1.16–1.74)	0.001	1.42 (1.16–1.75)	0.001
Overweight	279/4.57	1.32 (1.14–1.52)	<0.001	1.28 (1.10–1.48)	0.001	1.26 (1.09–1.46)	0.002	1.26 (1.09–1.47)	0.002
Obesity	71/4.72	1.35 (1.06–1.73)	0.017	1.23 (0.96–1.58)	0.103	1.21 (0.94–1.56)	0.132	1.22 (0.95–1.57)	0.116

Notes: BMI: body mass index; CHE: catastrophic health expenditure; cHR: crude hazard ratio; aHR: adjusted hazard ratio; CI: confident interval; Ref.: Reference. Model 1: Hazard ratio was adjusted for demographic characteristics, including gender, age group, education, marital status, and insurance. Model 2: Hazard ratio was additionally adjusted for health-related characteristics, including self-reported health, current smoking, drinking, chronic disease, and outpatient and inpatient services. Model 3: Hazard ratio was further adjusted for socioeconomic characteristics (residence, family economic level, family size, and socioeconomic development level), besides those factors included in model 2. * Per 1000 person-month.

**Table 3 nutrients-14-04014-t003:** Subgroup analyses for association of underweight and overweight with risk of CHE in female participants.

Subgroup	Normal Weight (Events/Objects)	Underweight			Overweight		
		Events/Objects	aHR (95%CI)	*p*-Value	Events/Objects	aHR (95%CI)	*p*-Value
All	519/1974	118/323	—	—	279/825	—	—
Age group							
16–39	109/662	25/120	1.33 (0.85–2.07)	0.213	35/181	1.07 (0.73–1.59)	0.724
40–49	127/575	24/70	1.60 (1.02–2.50)	0.042 *	70/262	1.25 (0.92–1.68)	0.150
50–59	125/400	24/47	1.69 (1.07–2.67)	0.025 *	99/238	1.45 (1.10–1.91)	0.008 *
≥60	158/337	45/86	1.23 (0.87–1.73)	0.240	75/144	1.30 (0.97–1.75)	0.078
Insurance							
None	74/291	17/51	1.41 (0.80–2.47)	0.232	39/132	1.20 (0.79–1.81)	0.402
UEBMI	48/254	3/18	0.90 (0.27–2.99)	0.863	28/127	1.00 (0.61–1.64)	0.994
URBMI	48/187	4/20	1.11 (0.38–3.29)	0.845	32/88	1.33 (0.79–2.26)	0.283
NRCMS	301/1029	81/189	1.52 (1.18–1.96)	0.001 *	151/407	1.24 (1.02–1.52)	0.033 *
Other	48/213	13/45	1.71 (0.87–3.33)	0.117	29/71	1.66 (0.99–2.79)	0.052
Self-reported health							
Good	175/823	34/122	1.16 (0.79–1.70)	0.443	104/327	1.44 (1.12–1.86)	0.005 *
Medium	199/791	43/126	1.53 (1.09–2.15)	0.014 *	108/335	1.23 (0.96–1.56)	0.095
Poor	145/360	41/75	1.53 (1.06–2.20)	0.022 *	67/163	1.09 (0.80–1.48)	0.573
Current smoking							
No	491/1876	113/307	1.47 (1.19–1.81)	<0.001 *	269/810	1.24 (1.07–1.45)	0.005 *
Yes	28/98	5/16	0.74 (0.21–2.60)	0.639	10/15	2.60 (0.96–7.10)	0.061
Current drinking							
No	504/1910	115/314	1.44 (1.17–1.77)	0.001 *	262/791	1.23 (1.06–1.44)	0.007 *
Yes	15/64	3/9	2.07 (0.40–10.80)	0.386	17/34	3.64 (1.40–9.47)	0.008 *
Chronic diseases							
No	416/1678	89/266	1.41 (1.11–1.78)	0.004 *	205/667	1.26 (1.06–1.50)	0.008 *
Yes	103/296	29/57	1.58 (1.03–2.43)	0.036 *	74/158	1.35 (0.99–1.85)	0.061
Outpatient services							
No	381/1511	71/239	1.17 (0.90–1.51)	0.239	202/641	1.19 (1.00–1.42)	0.053
Yes	138/463	47/84	2.17 (1.53–3.07)	<0.001 *	77/184	1.36 (1.02–1.82)	0.038 *
Inpatient services							
No	476/1830	110/295	1.45 (1.18–1.80)	0.001 *	261/768	1.30 (1.12–1.52)	0.001 *
Yes	43/144	8/28	0.98 (0.43–2.25)	0.969	18/57	0.63 (0.32–1.25)	0.189
Residence							
Urban	240/1048	35/138	1.17 (0.81–1.68)	0.404	162/508	1.24 (1.01–1.52)	0.044 *
Rural	279/926	83/185	1.60 (1.25–2.06)	<0.001 *	117/317	1.27 (1.02–1.58)	0.036 *
Socioeconomic development level							
Lowest	89/351	33/75	1.86 (1.23–2.82)	0.003 *	33/93	1.32 (0.87–2.00)	0.192
Lower	162/603	26/89	0.99 (0.65–1.52)	0.974	99/291	1.23 (0.95–1.60)	0.121
Higher	91/320	14/41	1.64 (0.92–2.94)	0.096	58/140	1.47 (1.04–2.07)	0.029 *
Highest	177/700	45/118	1.33 (0.94–1.9)	0.108	89/301	1.08 (0.84–1.41)	0.541
Family economic level							
Lowest	183/519	57/123	1.52 (1.12–2.07)	0.008 *	93/230	1.21 (0.93–1.58)	0.148
Lower	113/437	23/68	1.29 (0.80–2.08)	0.291	55/186	1.12 (0.80–1.56)	0.528
Higher	138/585	26/86	1.24 (0.81–1.91)	0.321	72/225	1.41 (1.05–1.90)	0.024 *
Highest	85/433	12/46	1.79 (0.95–3.37)	0.074	59/184	1.56 (1.10–2.22)	0.012 *

Notes: BMI: body mass index; CHE: catastrophic health expenditure; aHR: adjusted hazard ratio; CI: confident interval; UEBMI: urban employee basic medical insurance; URBMI: urban resident basic medical insurance; NRCMS: new rural cooperative medical scheme. * *p* < 0.05.

## Data Availability

All waves’ data from the CFPS can be downloaded from the CFPS official website (http://www.isss.pku.edu.cn/cfps/ accessed on 1 July 2022).

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
