# Peer review of "Association of Body Mass Index with Risk of Household Catastrophic Health Expenditure in China: A Population-Based Cohort Study"

_nutrients, 2022, doi:10.3390/nu14194014_

Round 1

Reviewer 1 Report

the theme is interesting with a linear research design. In my opinion, there would be a need for some improvements. At the end of the introduction, the research questions should be spelled out. In paragraph 2.2 procedures I would recommend adding, if available, scientific studies that used a similar analysis process, especially in the construction of the questionnaire. The other paragraphs of section 2 are clear and exhaustive. Sections 3 and 4 are also linear and legible. I would propose to move the last paragraph from line 316 to 323 in the conclusions. Furthermore, the usefulness of this type of study should be further developed in the conclusions

Reviewer 2 Report

1.     Page 2, line 67-68: Provide references for the statement “However, current evidence about association of BMI and 67 risk of CHE is insufficient”.

2.     Section 2.2: Provide references for the BMI cut-off values. 

3.     Explain how time to event was calculated for Cox regression models. 

4.     Discuss the generalizability of the findings to similar populations. 

5.     It is not clear why a sex-stratified analysis was undertaken. Please explain. 

6.     Comment on lost to follow-up rate at each wave of data collection. 

7.     Was multi-stage PPS taken in to account in Cox models?
